# Exposure to Oxidized Multi-Walled CNTs Can Lead to Oxidative Stress in the Asian Freshwater Clam *Corbicula fluminea* (Müller, 1774)

**DOI:** 10.3390/ijms242216122

**Published:** 2023-11-09

**Authors:** Antonio Cid-Samamed, Miguel Ángel Correa-Duarte, Andrea Mariño-López, Mário S. Diniz

**Affiliations:** 1Department of Physical Chemistry, Faculty of Sciences, University of Vigo, Campus de As Lagoas S/N, 32004 Ourense, Spain; 2Team NanoTech, Department of Physical Chemistry, University of Vigo, 36310 Vigo, Spain; macorrea@uvigo.es (M.Á.C.-D.); anmarino@uvigo.es (A.M.-L.); 3i4HB—Associate Laboratory Institute for Health and Bioeconomy, NOVA School of Science and Technology, NOVA University of Lisbon, 2819-516 Caparica, Portugal; mesd@fct.unl.pt; 4UCIBIO—Applied Molecular Biosciences Unit, NOVA School of Science and Technology, NOVA University of Lisbon, 2829-516 Caparica, Portugal

**Keywords:** oxidized multi-walled carbon nanotubes, biomarkers, oxidative stress enzymes, *Corbicula fluminea*

## Abstract

The increasing attention that carbon-based nanomaterials have attracted due to their distinctive properties makes them one of the most widely used nanomaterials for industrial purposes. However, their toxicity and environmental effects must be carefully studied, particularly regarding aquatic biota. The implications of these carbon-based nanomaterials on aquatic ecosystems, due to their potential entry or accidental release during manufacturing and treatment processes, need to be studied because their impacts upon living organisms are not fully understood. In this research work, the toxicity of oxidized multi-walled carbon nanotubes (Ox-MWCNTs) was measured using the freshwater bivalve (*Corbicula fluminea*) after exposure to different concentrations (0, 0.1, 0.2, and 0.5 mg·L^−1^ Ox-MWCNTs) for 14 days. The oxidized multi-walled carbon nanotubes were analyzed (pH, Raman microscopy, high-resolution electron microscopy, and dynamic light scattering), showing their properties and behavior (size, aggregation state, and structure) in water media. The antioxidant defenses in the organism’s digestive gland and gills were evaluated through measuring oxidative stress enzymes (glutathione-S-transferase, catalase, and superoxide dismutase), lipid peroxidation, and total ubiquitin. The results showed a concentration-dependent response of antioxidant enzymes (CAT and GST) in both tissues (gills and digestive glands) for all exposure periods in bivalves exposed to the different concentrations of oxidized multi-walled carbon nanotubes. Lipid peroxidation (MDA content) showed a variable response with the increase in oxidized multi-walled carbon nanotubes in the gills after 7 and 14 exposure days. Overall, after 14 days, there was an increase in total Ub compared to controls. Overall, the oxidative stress observed after the exposure of *Corbicula fluminea* to oxidized multi-walled carbon nanotubes indicates that the discharge of these nanomaterials into aquatic ecosystems can affect the biota as well as potentially accumulate in the trophic chain, and may even put human health at risk if they ingest contaminated animals.

## 1. Introduction

The use of carbon-based nanomaterials (CBNMs) has increased in recent decades due to their distinctive thermal, optical, mechanical, and electromagnetic properties [1,2]. The release of CBNMs is recognized worldwide as an emerging problem, with most of these nanomaterials ending up in different environmental compartments [3]. The accidental release of nanomaterials into aquatic environments is a global concern, as its effects and consequences on aquatic ecosystems are still unclear. The release of CBNMs, mainly through industrial and urban wastewaters, constitutes a potential risk for the environment and inhabiting species [4,5]. It can pose a significant threat to human health since its entry into aquatic systems can lead to effects on the trophic/food chain, such as bioaccumulation and biomagnification. Thus, multiple pathways affecting human health may arise from exposure to CBNMs used in commercial products. 

Nanoparticles are usually defined as having at least one dimension and size up to 100 nm [6]. Carbon nanotubes (CNTs) are hollow cylinders of graphene that range from microns to millimeters in length and can be classified into single-walled (SWCNTs) with a diameter of 0.7 to 3 nm and multi-walled (MWCNTs) with a diameter of 10 to 25 nm [7,8]. CNTs are engineered nanomaterials with wide-ranging core structures and surface modifications that change their physical and chemical characteristics to improve their suitability for diverse manufacturing uses [9]. Due to their non-biodegradable and hydrophobic properties, which make their dispersion in water challenging [10], CNTs can be accumulated by aquatic biota through the body surface and the respiratory and digestive systems [11]. The bioaccumulation of CNTs has been previously reported in aquatic invertebrates and bivalves [12,13,14]. However, the colloidal behavior of CNTs relies on their chemical attributes, such as size, aggregation rate, and sorption phenomena. On the other hand, abiotic factors such as pH, temperature, and the aquatic system’s ionic strength are ruled by this behavior. All these factors affect the behavior of CBNMs and play a key role in their chemical reactivity; surface phenomena; aggregation rate, which influences their location in the aquatic system; fate; bioavailability; and, finally, their toxicity. However, the toxicological impacts of CNTs, but more generally of CBNMs, depend on several issues, comprising a complex interaction among particle qualities (e.g., diameter, shape, surficial charge, and surficial chemistry), the chemical nature of the materials, concentration, exposure time, medium exposure arrangement, and the route of particle administration [15]. CNTs willingly aggregate in solution [16].

Therefore, these CBNMs can be dispersed through functionalization that is attained via chemical alteration, such as the esterification and amidation of nanotube-bound carboxylic acids [17]. Functionalization breaks nanotube packages, essential to solubility [11], and functional groups’ occurrence on the nanotubes surface, increasing their dispersibility [18]. Precisely, to disperse CNTs in water, the chemical modification of CNTs through adding polar groups, such as a carboxyl group (-COOH), is one of the most common forms of functionalization to improve dispersibility [15]. Another study concluded that functional groups on the sidewall of MWCNTs meaningfully amplified their dispersibility [8].

The remarkable development of nanotechnology became an environmental concern because the risk of accidental release into water sources required the study of nanomaterials’ effects on the environment and, therefore, on aquatic biota [19].

Carbon-based nanomaterials have attracted widespread attention during the last decades; given their potential applications, these nanomaterials acquired a wide range of modifications. A particular modification is focused on making them more easily dispersed in water. Compared to new and organic-soluble CNMs, water-soluble CNTs are more likely to enter the human body and aquatic environment [5]. There are two main approaches to enhancing the dispersibility of CNTs in aqueous media. The first is the chemical functionalization of CNTs through adding polar groups, such as carboxyl groups, and the other is the physical functionalization of CNTs through the adsorption of polymers and tensioactives for reducing the van der Waals interaction [18].

Many studies have reported NPs’ effects on living organisms, including various aquatic species such as bivalves [20,21,22,23,24,25,26,27]. Nevertheless, the existing scientific literature on the impacts of oxidized MWCNTs is limited. To our knowledge, this is the first study in which Ox-MWCNTs toxicity was evaluated in *C. fluminea*, a freshwater filter-feeding bivalve. This benthic bivalve has a great potential to link nutrient energy fluxes among sediment and water columns, which has been previously employed as a delicate biological model to evaluate the toxicity and bioaccumulation of various contaminants [26,27,28].

Aquatic bivalves have been reported to be used as ecological indicators. Thus, the Asian freshwater clam, *C. fluminea,* has been selected as a biological model for the present work due to its worldwide spread as an invasive species, and as a benthonic filter-feeding bivalve, it can provide the measure of pollutant effect level on the aquatic source where they inhabit. Additionally, this freshwater bivalve has been used to assess the impact of nanomaterials on aquatic biota [8,25,29]. Therefore, they can indicate the ecosystem’s pollution, such as *C. fluminea,* which can be considered as a contamination pointer because of its capacity to bioaccumulate and tolerate some pollutants, as was previously reported by [29,30,31].

The present study aimed to use *C. fluminea*, a freshwater bivalve, as a biological model to assess different concentrations of surface-modified MWCNTs, mainly the toxicity of oxidized multi-walled carbon nanotubes (Ox-MWCNTs). The surface modification of MWCNTs alters their water solubility, so their accidental release into aquatic systems and subsequent effects in aquatic biota are of significant concern. It must be carefully determined, particularly in aquatic environments, where Ox-MWCNTs can ultimately accumulate [26,27,32,33]. For the first time, this study evaluated the survival, lipid peroxidation response (malondialdehyde levels), some stress oxidative enzymes (glutathione-S-transferase and catalase), superoxide dismutase, and total ubiquitin levels in animals exposed to different concentrations of Ox-MWCNTs.

This study hypothesizes that exposure, even at lower concentrations, will produce changes in antioxidant defenses and cellular damage biomarkers, suggesting that Ox-MWCNTs may cause oxidative stress and cell damage in this species.

## 2. Results

### 2.1. Characterization of Ox-MWCNTs

The characterization of Ox-MWCNTs suspensions corroborated the information supplied by the producer before the functionalization process, showing an average particle size of 9.5 nm in diameter and 1.5 µm in length. This characterization is often mandatory in nanotoxicology studies to understand the behavior of the CBNMs used during exposure assays. The characterization of the oxidized-MWCNTs suspensions was accomplished via Raman analyses and transmission electron microscopy (TEM) analysis (Figure 1 and Figure 2). 

### 2.2. Mortality Rate

After 7 days, 10% mortality was observed in animals exposed to 0.1 mg·L^−1^ of Ox-MWCNTs, and after 7 and 14 days in animals exposed to 0.2 mg·L^−1^ of Ox-MWCNTs. 

### 2.3. Biochemical Analyses

#### 2.3.1. Catalase (CAT)

The CAT activity was measured in the digestive glands (DGs) and gills (Gs) of *C. fluminea* subjected to different concentrations of Ox-MWCNTs·L^−1^ and different exposure periods (7 and 14 days). 

The catalase results are shown in Figure 3a,b, with the lowest average value (0.05 nmol min^−1^ mg^−1^ of total protein) being measured in the gills of control animals (0 mg·L^−1^ of Ox-MWCNTs) after 14 days, while the highest level of catalase activity (1.96 nmol min^−1^ mg^−1^ total protein) was measured in organisms’ DGs exposed to 0.1 mg·L^−1^ of Ox-MWCNTs after 7 exposure days. A significant increase (*p* < 0.05) was found in the organism’s gills exposed to Ox-MWCNTs concentrations of 0.1 mg·L^−1^ after 7 days of exposure. No significant (*p* > 0.05) changes were detected in both organs of the animals exposed to 0.5 Ox-MWCNTs mg·L^−1^ after 7 and 14 days of exposure compared to controls. Data are also presented in Appendix A.

#### 2.3.2. Glutathione-S-Transferase (GST) 

The results of GST activity determined in gills and DG of *C. fluminea* are shown in Figure 4a,b. Data are also presented in Appendix A. The results show an enhancement of GST activities in both tissues (digestive glands and gills) of animals exposed to Ox-MWCNTs for 14 days, whereas in animals exposed to Ox-MWCNTs for 7 days, both tissues low GST activities were observed for all concentrations tested compared to respective controls. Regarding the gills, the highest activities of GST were recorded in animals exposed to 0.2 mg of OX-MWCNTs·L^−1^ (6.25 nmol·min^−1^·mg^−1^ total protein) after 14 days of exposure, whereas the lowest average value measured in the gills was found in organisms exposed to 0.1 mg of OX-MWCNTs·L^−1^ (1.10 nmol·min^−1^·mg^−1^ total protein) after 7 days of exposure. In gills, differences were found in the gills and DG of animals exposed to 0.1 mg of OX-MWCNTs·L^−1^ compared to respective controls after 7 days of exposure. Following 14 days of exposure, significant differences were found between controls and all tested concentrations. In addition, significant differences were also found between exposure periods for all tested concentrations except the control (0 mg of OX-MWCNTs·L^−1^). 

Regarding the DGs, the highest activities of GST were registered in organisms exposed to 0.1 mg of OX-MWCNTs·L^−1^ (10.76 nmol·min^−1^·mg^−1^ total protein) after 14 exposure days, whereas the lowest average value measured in the DGs was found in animals exposed to 0.2 mg of OX-MWCNTs·L^−1^ (4.09 nmol·min^−1^·mg^−1^ total protein) after 7 exposure days. Regarding DG, significant differences were found after 7 days of exposure to 0.2 mg of OX-MWCNTs·L^−1^. Moreover, significant differences were also found after 14 days of exposure to 0.1 and 0.5 mg of OX-MWCNTs·L^−1^. Additionally, significant differences between exposure periods were found for all tested concentrations except for the controls (0 mg of OX-MWCNTs·L^−1^).

#### 2.3.3. Lipoperoxidation (LPO)

The LPO (MDA content) results measured in the gills and DGs of *C. fluminea* are shown in Figure 5a,b. Data are also presented in Appendix A. The lowest average value was measured in the gills of organisms exposed to 0.2 mg of Ox-MWCNTs·L^−1^ (0.25 nmol·mg^−1^ total protein), and the highest average value (1.89 nmol mg^−1^·total protein) was measured in the DGs of animals exposed to 0.1 mg·L^−1^ of Ox-MWCNTs after 14 days of exposure. The results show an increasing trend after seven days of exposure in the gills according to the different tested Ox-MWCNTs concentrations, except for bivalves exposed to 0.5 mg of Ox-MWCNTs·L^−1^. However, after 14 days of exposure, a decrease in MDA content was found compared with controls.

Generally, lipid peroxidation showed a variable response according to the increasing concentrations of Ox-MWCNTs in gills after 7 and 14 days of exposure. Concerning DGs, the lowest tested concentration (0.1 mg Ox-MWCNTs·L^−1^) showed the highest level of MDA after 14 days of exposure.

An upsurge in MDA content was measured in the animals’ gills after 7 days of exposure to 0.1 and 0.2 mg of Ox-MWCNTs·L^−1^ compared to controls. Significantly low MDA content (*p* < 0.05), measured in the animals’ gills, after 14 days of exposure, was observed for all concentrations tested. Additionally, all these concentrations tested showed significant differences compared to the respective controls. In addition, significative differences were also detected for all tested concentrations between exposure periods. 

Concerning DGs, a slight increase was found after 7 days of exposure to 0.2 and 0.5 of Ox-MWCNTs·L^−1^ compared to respective controls, and no significative differences were found between the different concentrations tested. Following 14 exposure days, a rise was found in the DGs of animals exposed to 0.1 of Ox-MWCNTs·L^−1^ in comparison to the respective controls, and lower values of MDA content were measured in the DGs of animals exposed to 0.2 and 0.5 of Ox-MWCNTs·L^−1^, showing significant differences compared to controls for these two higher tested concentrations. Comparing the effects of different exposure periods (*p* < 0.05), significant differences were detected among the controls and the DGs of animals exposed to 0.1 of Ox-MWCNTs·L^−1^.

#### 2.3.4. SOD

SOD activity was determined in the gills and DGs of *C. fluminea*, and results are shown in Figure 6a,b. Data are also presented in Appendix A. The lowest average value of SOD activity was 3.32 Units·mg^−1^ total protein, observed in DGs exposed to 0.5 mg Ox-MWCNTs·L^−1^ after 7 exposure days, and the maximum average value was 5.65 Units.mg^−1^ total protein found in gills exposed to 0 mg of Ox-MWCNTs·L^−1^ for 7 days.

SOD activities did not show great variability but showed some significant differences (*p* < 0.05) found in each concentration of Ox-MWCNTs tested compared with respective controls in gills and between exposure periods for controls and 0.1 and 0.5 mg of ox-MWCNTs·L^−1^. Significant differences were found only in the DGs of animals exposed to 0.1 mg Ox-MWCNTs·L^−1^. Additionally, significant differences (*p* < 0.05) were found amid exposure periods for controls and 0.1 and and 0.5 mg of Ox-MWCNTs·L^−1^ in gills; and for controls, 0.2 mg Ox-MWCNTs·L^−1^ was found in the digestive glands. 

#### 2.3.5. Total Ubiquitin (Ub)

The total ubiquitin results determined in the gills and DGs of *C. fluminea* are shown in Figure 7a,b. Data are also presented in Appendix A.

The lowest average value of total ubiquitin was found in the DGs (0.0045 ± 0.0018 mg·mg^−1^ total protein) of bivalves exposed to 0.2 mg of OX-MWCNTs·L^−1^ after 14 days of exposure, while the highest average value (0.0135 ± 0.0064 mg·mg^−1^ of total protein) was found in the gills of organisms exposed to 0.1 mg of Ox-MWCNTs·L^−1^ after seven days of exposure.

Total ubiquitin showed a variable response with the increase in Ox-MWCNTs concentrations tested for both organs analyzed and all exposure periods assessed. 

After 7 days of exposure, only the gills of animals exposed to 0.2 mg Ox-MWCNTs·L^−1^ presented significant differences (*p* < 0.05) compared with controls. Moreover, significant differences were found between exposure periods for controls and 0.1 mg Ox-MWCNTs·L^−1^ for both organs.

Concerning DGs, after 7 days of exposure, all tested concentrations showed lower levels of total Ub, and no significant differences were found compared with the respective controls. After 14 days of exposure, only 0.1 mg Ox-MWCNTs·L^−1^ was higher than the respective control, and there were no significant differences. Significant differences (*p* < 0.05) were found between exposure periods, controls, and 0.1 and 0.5 mg Ox-MWCNTs·L^−1^.

#### 2.3.6. Correlation Analyses

Figure 8 shows the correlation results for biomarkers analyzed for both organs and exposure periods. A correlation matrix drawn from the activities of biomarkers analyzed shows a higher correlation for DGs than for Gs, with the exposure period revealed as a critical parameter for positively and negatively related biomarkers, and different antioxidant defenses show a compensatory relationship.

## 3. Discussion

The CNT concentrations tested in the present study were selected based on CNT concentrations measured in wastewaters [5]. Although environmentally relevant concentrations were estimated, this study offers relevant knowledge on the effects of Ox-MWCNTs in aquatic media. On the other hand, there is lack of information concerning concentrations of nanomaterials near industrial wastewater effluents or wastewater treatment plants [9].

In the present study, exposure to lower concentrations also produced changes in oxidative stress enzymes (e.g., GST, CAT, and SOD), suggesting that Ox-MWCNTs may cause oxidative stress in *C. fluminea*. However, CAT activities in both tissues were higher after 7 days of exposure than after 14 days, suggesting a more intense stress response in this exposure period. Since CAT activity was higher in organisms exposed to 0.1 mg Ox-MWCNTs·L^−1^, for both tissues, a decline in the CAT activity was observed. 

However, GST activity showed a more significant effect after 14 days than after 7 days of exposure. The GST results suggest that higher activities were found for all tested concentrations after 14 days of exposure, showing significant differences compared to the 7-day exposure period. The results suggest that these filter-feeding benthic bivalves may internalize the OX-MWCNTs, promoting their bioaccumulation in these organisms and causing toxicity. The LPO results (MDA content) showed opposite results in the organs analyzed: an increase was found after 7 days in the gills, and in DGs after 14 days of exposure, possibly related to the tissue specialization of each organ. While the gills are the respiratory system and interact directly with the aquatic environment, the digestive gland (DG) is the metabolic and excretion organ. The capture of nanoparticle aggregates accompanied by oxidative stress was also reported by previous researchers for gill epithelial cells [34,35]. Nanoparticles in general and CBNMs in contact with digestive cells are known to cause lipid peroxidation and a decrease in the stability of lysosomal membranes [31,36]. It is known that during stressful conditions, excessive ROS generated by organisms are not effectively removed by oxidative stress defense mechanisms, including antioxidant enzymes, which can result in lipid peroxidation (LPO) and other cellular damages [8]. It has been shown that an important mechanism of toxicity of NMs is oxidative stress, associated with an increase in reactive radicals that can disrupt the balance between antioxidant and oxidative damage, resulting in significant sublethal toxicity to organisms [31,37,38,39,40,41,42,43]. Cellular responses following exposure to MWCNTs are commonly marked by mitochondrial permeability, ROS production, lysosomal membrane destabilization (LMD), and ultimate cell death via the apoptotic pathway. ROS production seems to precede the other cellular responses, and enhanced ROS production has been associated with LMD [44,45]. The dose-dependent cytotoxicity produced by MWCNTs may be closely linked to increased oxidative stress [46].

Concerning SOD, we observed that exposure to Ox-MWCNTs caused a decrease in SOD activity; thus, it can be suggested that other defense mechanisms against oxidative stress were also triggered. Living organisms show diverse defense mechanisms to counter the ROS excess, typically enzymes (e.g., SOD, CAT) and other compounds such as ubiquinols, vitamin A, and carotenes. Once the capability of the antioxidant system is surpassed, adverse outcomes upon the organisms arise [47,48].

CAT and SOD are the primary enzymes acting to remove ROS [25,47,49]. The results generally show sub-lethal effects in bivalves exposed to the different Ox-MWCNT concentrations. Though there was a trend for SOD to rise consistent with the concentrations of Ox-MWCNTs tested, no significant differences were detected. Nevertheless, CAT activities exhibited a general increase, but not significant, in clams exposed to Ox-MWCNTs, denoting that CAT counteracts H_2_O_2_ excess. We can postulate that GST additionally fights oxidative stress produced by the exposure to Ox-MWCNTs since this enzyme is known to aid in the oxidative stress fight [50]. 

Ubiquitins are intracellular proteins that target unwanted proteins for proteolytic degradation [51]. Thus, regarding the DG of *C. fluminea* exposed to 0.1 and 0.2 mg Ox-MWCNTs·L^−1^, no significant differences were observed compared to controls, although there is a trend towards a decrease in total Ubiquitin. However, a decrease in total Ubiquitin was found in animals exposed to 0.5 mg Ox-MWCNTs·L^−1^ after 14 days. The Ubiquitin-dependent proteasomal degradation machinery is sensitive to oxidative stress, beginning to decrease its activity at high H_2_O_2_ concentrations [52]. Under persistent oxidative stress, the ubiquitination system is inactivated, and ubiquitin levels tend to decrease [28]. Instead, in these cases, proteasome machinery, independent of ubiquitin, requires less energy, is less sensitive to oxidative stress, is more active, and ensures the proper degradation of damaged proteins [53].

Furthermore, the variability observed in controls for some biomarkers (e.g., CAT, LPO, SOD, and ubiquitin) can be ascribed to species variability, genetic heritage, sex, or other environmental conditions that the organisms may have been exposed to [54]. 

Regarding the mortality rate (10%) registered in organisms exposed to 0.1 and 0.2 mg Ox-MWCNTs·L^−1^ after 7 days and in organisms exposed to 0.2 mg Ox-MWCNTs·L^−1^ for 14 days of exposure, it is considered low and can be explained by the fact that some individuals may have some pathological condition or are older and thus be more susceptible. 

The carboxylation of MWCNTs through adding polar groups, for example, carboxyl groups (–COOH), has been shown to have more amorphous carbon fragments due to increased carbon oxidation, and these amorphous portions may provoke greater toxicity in biological systems compared to non-functionalized CNTs [49].

The moderate sub-lethal effects determined in the current study can be attributed to various factors such as the colloidal aggregation behavior of Ox-MWCNTs in freshwater media, the dose, the functionalized group, the chemical and physical properties of the Ox-MWCNTs, and the ability of the organisms to fight oxidative stress. However, since some studies show opposite results due to the use of different concentrations and biological models, further studies are needed to improve the understanding of the impacts of Ox-MWCNTs on aquatic biota.

The results provide information about the ecosystems health, including aquatic biota, if contamination with CNTs occurs. Nonetheless, if these bivalves are consumed by humans, then some effects can occur, mainly in lung epithelial cells [55] but also in the cardiovascular system [56] or the immune system [57], among other effects. However, the knowledge on its toxicity determinants remains to be clarified [58], and, thus, a better understanding is needed [59].

## 4. Materials and Methods

### 4.1. Functionalization and Characterization of Ox-MWCNTs

Nanocyl S.A., Sambreville, Belgium, provided MWCNTs. They were synthesized via the catalytic chemical vapor deposition (CCVD) process and purified up to 95 wt.%, with an average particle size of 9.5 nm in diameter and 1.5 µm in length. The characterization of the oxidized-MWCNTs suspensions was performed via transmission electron microscopy (TEM) analysis. Afterward, MWCNTs were pre-treated via sonication in a mixture of acetone/ethanol (1:1) to remove organic materials and washed with ultrapure water using three centrifugation cycles (9000 rpm, 20 min). Then, MWCNTs were frozen at −20 °C overnight and lyophilized for 48 h (0.8 mbar, −80 °C). Afterward, 100 mg of the pre-treated MWCNTs was oxidized via sonication in 100 mL of an H_2_SO_4_/HNO_3_ (3:1) mixture for 15 min using an ultrasonic probe (20 W) and, after that, sonicated in an ultrasonic bath (Bandelin Sonopuls, 200 W—working at 25% power, 50 W—and 20 kHz) at room temperature for 4 h. The sample was washed with 0.1 M NaOH aqueous solution using four re-dispersion cycles (9000 rpm, 30 min). Once the pH stabilized at 10, the sample was sonicated in an ice bath using the tip sonicator for 2 h and washed with water using three centrifugation cycles (9000 rpm, 8 h). A dispersion of oxidized MWCNTs (0.75 mg/mL) was obtained [60].

The samples were dispersed in water, dried on a carbon membrane coated grid, and examined via TEM and HRTEM (high-resolution transmission electron microscopy); TEM was performed using a JEOL JEM1010 TEM (JEOL, Tokyo, Japan) working at an accelerating voltage = 100 kV, whereas HRTEM was carried out on a JEOL JEM2010F HRTEM (JEOL, Tokyo, Japan) operating at 200 kV [61]. Then, plates were scanned, and image analysis was conducted using Image J software (https://imagej.softonic.com/mac, accessed on 18 October 2023, NIH, Bethesda, MD, USA) after optimizing the parameters (area of analysis = 4–200 nm^2^; circularity = 0.4–1).

Raman spectra were measured with a Renishaw InVia system using a 785 nm laser. The laser beam was focused on the sample using a 50× objective, with power at the sample of 2.87 mW and acquisition times of 10 s. 

### 4.2. Experiment Setup

For bioassays, stock solutions of Ox-MWCNT suspensions (4.52 mg·L^−1^) were prepared in ultra-pure filtered water (millipore–0.2 µm filters) and then sonicated in an ultrasound bath (J.P. Selecta, Abrera, Spain) at 60 kHz for 24 h.

The freshwater bivalves, *C. fluminea*, were manually gathered in the Tagus River (Salvaterra de Magos, Portugal) and acclimatized for fourteen days under experimental settings in a closed, continuous-flow system (200 L volume capacity) with de-chlorinated and filtered tap water. Bivalves were daily fed ad libitum with suspended dehydrated microalgae (Chlorella sp.) (DibaQ, Segovia, Spain). The pH, ammonia, and temperature of the water were checked daily. The exposure test was performed in semi-static mode and in triplicate, at 20 ± 1 °C and pH 7.09 ± 0.07, with a 12 h light:12 h dark photoperiod with continuous aeration (>6 mg·L^−1^).

The bivalves (*n* = 96) weighed 18.4 ± 4.6 g (edible part) and had a shell length of 3.5 ± 0.3. Then, 12 bivalves, for each concentration, were randomly distributed among eight polyvinyl tanks of 20 L volume capacity and exposed to different nominal concentrations of Ox-MWCNTs (0.1, 0.2, and 0.5 mg·L^−1^), freshly prepared and used through diluting the stock Ox-MWCNTs suspension solution in Milli-Q water, submitted to sonication in an ultrasound bath for 20 min at 25 °C to attain the required final concentrations in each assay tank (volume capacity of 6 L). A supplementary tank with de-chlorinated and filtered tap water was used as a control. Previous experiments were carried out showing no influence of the composition of the tanks in the experiments.

Prior to the beginning of the trials (T0), six bivalves were collected and assayed for stress oxidative enzymes, LPO, and Ubiquitin. Six animals of each tested concentration were collected after 7 and 14 days of exposure. The effect of pollutants or nanometric stressors has been reported to be well-known in short-term exposure experiments [62]. The bivalve’s organs were removed (digestive gland and gills) using a forceps and a scalpel. For every sampling period and concentration tested, six individuals were sampled and both organs were removed, placed in microtubes (1.5 mL), and stored (−80 °C) until later analyses. 

The concentrations tested (0.2 mg Ox-MWCNTs·L^−1^) were selected based on data from the scientific literature [5,9,63,64], whereas the highest concentrations were tested to assess the effects and compare with the lower exposure concentrations.

### 4.3. Biochemical Analysis

Samples were homogenized in 3 mL of phosphate-buffered saline solution (PBS) (0.14 M NaCl, 2.7 mM KCl, 8.1 mM Na_2_HPO_4_, 1.47 mM KH_2_PO_4_; pH 7.4) and then transferred to microtubes (1.5 mL volume), centrifugated at 10,000× *g* (15 min; 4 °C), and stored at −80 °C for later analyses. The total protein mass (mg) was measured to normalize the biomarker results [65].

#### 4.3.1. Stress Oxidative Enzymes

##### Glutathione-S-Transferase (GST)

Glutathione-S-transferase (GST) activity was determined through following the method described by Habig et al., 1974 [66], and optimized for 96-well microplates. A mixing solution was prepared in Dulbecco’s phosphate-buffered saline solution (200 mM reduced L-glutathione prepared in ultrapure water (Sigma-Aldrich) plus 100 mM CDNB prepared in ethanol (96%) (Sigma-Aldrich)). Then, 20 µL of sample or GST standard were added into each well of a 96-well microplate (Greiner Bio-One GmbH, Frickenhausen, Germany), followed by adding 180 µL of the mixed solution to each microplate well. Equine liver GST was used as a positive control (Sigma–Aldrich). The enzyme activities were measured through recording the absorbance (at λ = 340 nm) for 6 min, using a microplate reader (Bio-Rad Benchmark, Hercules, CA, USA), and the activity was calculated using a CDNB extinction coefficient of 0.0053 mM. The results are expressed in relation to the sample’s total protein concentration.

##### Catalase

Catalase (CAT) activity was measured as previously described by Johansson and Borg, 1988 [67], after adapting to 96-well microplates. This method is based on the enzyme’s reaction with methanol in the presence of an appropriate concentration of hydrogen peroxide (H_2_O_2_). The formaldehyde produced is determined using Purpald (4-amino-3-hydrazino-5-mercapto-1,2,4-triazole) as a chromogen. Concisely, 20 μL of each sample or standard, 100 μL of assay buffer (100 mM potassium phosphate), and 30 μL of methanol (Scharlab, Barcelona, Spain) were added to each well of a 96-well microplate (Greiner Bio-One GmbH, Frickenhausen, Germany). Next, the reaction was started through adding 20 μL of H_2_O_2_ (0.035 M, 30%, Sigma-Aldrich) to each well, and the microplate was incubated for 20 min on a shaker. Subsequently, 30 μL of KOH (10 M, Chem-Lab, Zedelgem, Belgium) and 30 μL of Purpald (34.2 mM) in HCl (0.5 M, Sigma-Aldrich) were added into each well and incubated for 10 min in shaker at room temperature. Next, 10 μL of KIO_4_ (65.2 mM) in KOH (0.5 M, Sigma-Aldrich, St. Louis, MO, USA) was added to all wells and incubated for 5 min. Then, the absorbance was measured at 540 nm in a microplate reader (Bio-Rad, Benchmark, USA). A calibration curve was prepared from a formaldehyde (4.25 mM, Sigma-Aldrich) stock solution to obtain concentrations ranging from 0 to 75 μM, which served to determine the formaldehyde concentration in the samples. Results are presented in relation to the total protein concentration (nmol/mg total protein).

##### Superoxide Dismutase

Superoxide dismutase (SOD) was measured as first described by McCord and Fridovich, 1969 [68]. A reaction mixture was prepared, containing 11.5 mL of distilled H_2_O, 12.5 mL of PBS (Sigma-Aldrich), 0.5 mL and EDTA (Honey Well Riedel-de-Haën, Seelze, Germany), 0.5 mL of Cyt c, and 2.5 mL of Xanthine. The mixture was then adjusted to pH 7.8. To measure the inhibition percentage of SOD, each of the 96 wells of the plate (Greiner, Bio-One GmbH, Frickenhausen, Germany) contained 280 µL of the mixture previously prepared, 10 µL of XOD (Sigma-Aldrich), 10 µL of sample, and 10 µL of H_2_O. Blanks were also prepared through adding 280 µL of the mixture, 10 µL of XOD, and 10 µL of H_2_O.

The absorbance was measured at 540 nm using a microplate reader (Bio-Rad, Benchmark, USA) every 5 min for 15 min. The inhibition percentage of SOD is expressed in Units·mg^−1^ total protein. 

#### 4.3.2. Biomarkers of Cellular and Protein Damage

##### Lipid Peroxidation Assay

The lipid peroxidation assay followed the thiobarbituric acid reactive substances (TBARS) assay [69]. In brief, 5 µL of each sample was added to 45 µL of 50 mM monobasic sodium phosphate buffer in microtubes (1.5 mL). Then, 12.5 µL of SDS (at 8.1%), 93.5 µL of trichloroacetic acid (20%, pH 3.5), and 93.5 µL of thiobarbituric acid (1%) were added to each microtube. To this mixture, 50.5 µL of ultrapure water (MQ-grade) was added in each microtube and vortexed for 30 s. The microtubes’ caps were perforated with a pin and then introduced in boiling water for 10 min. They were then placed on ice for a few minutes to cool. Afterwards, 62.5 mL of ultrapure water (MQ-grade) was added to each microtube, followed by 312.5 mL of n-butanol pyridine (15:1, *v*/*v*). Then, microtubes were centrifuged at 5000× *g* for 5 min. Duplicates of 150 µL of each reaction’s supernatants were placed into the wells of a 96-well microplate (Greiner, Bio-One GmbH, Frickenhausen, Germany), and the absorbance was measured at λ = 530 nm using a microplate reader (Bio-Rad Benchmark, Hercules, CA, USA). To determine lipid peroxidation, an eight-point calibration curve (0–0.1 µM) was built using malondialdehyde bis (dimethyl acetal) (MDA) as the standard (Merck, Darmstadt, Germany). The results were expressed in relation to total protein concentration (nmol/mg total protein).

##### Total Ubiquitin

The quantification of total ubiquitin was conducted using a direct ELISA [70]. Triplicates of 50 μL were added into the microplate wells and left to incubate overnight at 4 °C. Afterwards, the microplates were washed (3×) in PBS with 0.05% Tween-20. Then, 200 μL of a blocking solution containing 1% BSA (Bovine Serum Albumin, Sigma-Aldrich, USA) in PBS was added to each microplate well. Then, the microplates were incubated at 37 °C for 90 min. Then, the microplates were washed once again, and the primary antibody (Ub P4D1, sc-8017, HRP conjugate, Santa Cruz, CA, USA) was prepared to 0.5 μg.mL^−1^, in 1% BSA, and 50 μL was added into the microplates’ wells. Then, the microplates were allowed to incubate for 90 min at 37 °C. After the new washing step, 50 μL of the substrate (TMB/E, Merck Millipore, Temecula, CA, USA) was pipetted to each well and then allowed to incubate for 15 min at room temperature. Then, 50 μL of stop solution (1 M HCl) was added to each well, and the absorbance was read at 450 nm in a microplate reader (BIO-RAD, Benchmark, USA).

For quantification purposes, a calibration curve was built using serial dilutions of purified ubiquitin (UbpBio, E-1100, Dallas, TX, USA) ranging from 0 to 0.8 μg.mL^−1^. Results were expressed in relation to total protein concentration (μg/mg total protein).

### 4.4. Statistical Analysis

The homogeneity of variances and normal distribution were checked using Shapiro–Wilk and Levene’s tests. After failing to verify these statistical assumptions, the non-parametric Kruskal–Wallis test was conducted to identify significant differences between controls and treatments. The significance level was fixed at 5%. The statistical analysis was performed employing the software Statistica 8.0 (StatSoft Inc., Tulsa, OK, USA, 2007). Correlation analyses (Pearson r correlation coefficient) were carried out using the software Graphpad-Prism 9 (Boston, MA, USA).

## 5. Conclusions

Considering the main results obtained in this study, the results indicate a relationship according to the concentration of Ox-MWCNTs tested, and the response of some stress oxidative enzymes (CAT, LPO, and SOD) in bivalves’ gills, DGs, and total ubiquitin. Consequently, exposure to Ox-MWCNTs causes an oxidative stress response and a rise in lipid peroxidation, which may suggest cell damage, observed in the gills after 7-day exposure to 0.1 and 0.2 mg Ox-MWCNTs·L^−1^ and in the DGs after 14-day of exposure to 0.1 mg Ox-MWCNTs·L^−1^. 

Furthermore, considering the widespread attention attracted by nanotechnology and industrial uses of CBNMs and the resulting potential discharge during the lifetime of nanomaterials (i.e., different states of nanomaterials, from production to treatment processes to achieve deliverable marketable products) to aquatic environments, the present study reports on novel and useful information concerning the possible risk to the environment and biota. Thus, it also provides relevant information on the potential entry and accumulation of Ox-MWCNTs in aquatic biota, which may be consumed by humans.

Given our results, we can hypothesize that the more intense responses observed at lower concentrations for some biomarkers are due to their higher dispersion in the water. Compared to higher concentrations, the aggregation of CBNMs and deposition to the bottom occurred, with Ox-MWCNTs being less bioavailable. 

This work provides valuable information on oxidative stress caused by exposure to Ox-MWCNTs, which may be due to their great potential for dispersion in water media.

In addition, it is crucial to elucidate the mechanisms and potential effects of Ox-MWCNTs’ toxicity to aquatic organisms. Consequently, additional studies are required to improve the understanding of the main sources, pathways, and behavior of these CBNMs in aquatic environments.

## Figures and Tables

**Figure 1 ijms-24-16122-f001:**
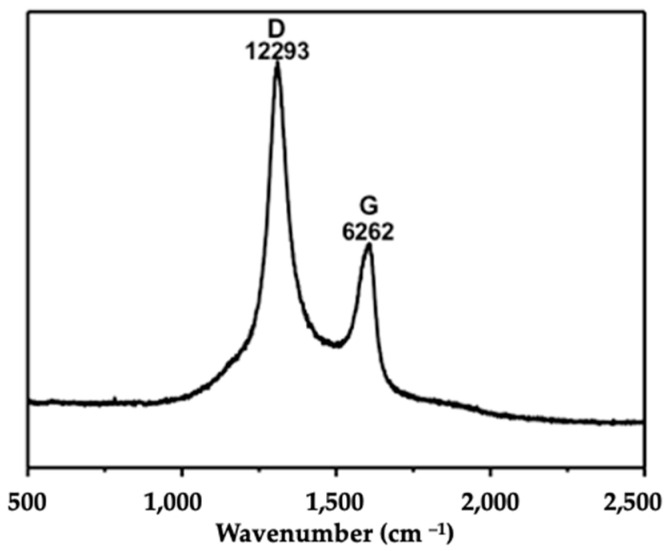
Representative Raman microscopy spectrum.

**Figure 2 ijms-24-16122-f002:**
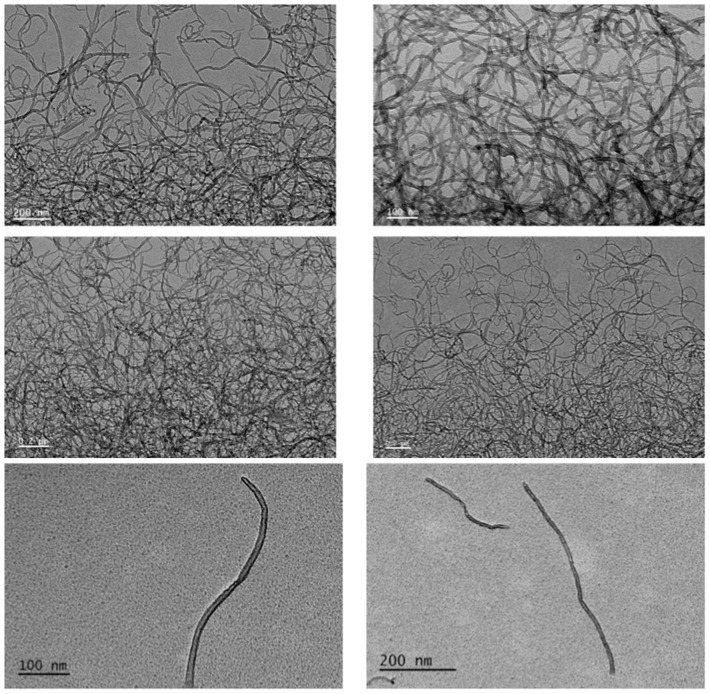
Representative TEM and HRTEM micrographs of the Ox-MWCNTs suspensions (scale bars: 0.2 μm, 200, 100 nm).

**Figure 3 ijms-24-16122-f003:**
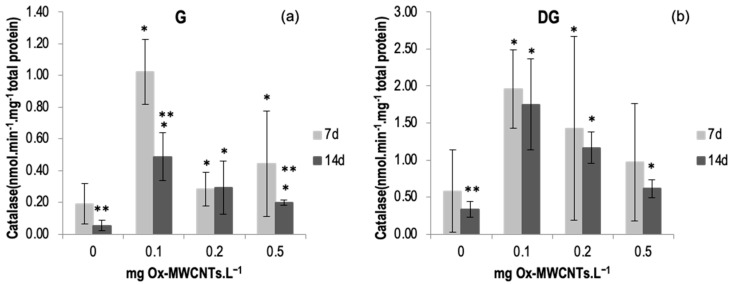
(**a**) Catalase activities measured in gills and (**b**) in digestive glands of *C. fluminea* exposed to different concentrations of Ox-MWCNTs. Legend: significant differences from controls (*) and between exposure periods (**).

**Figure 4 ijms-24-16122-f004:**
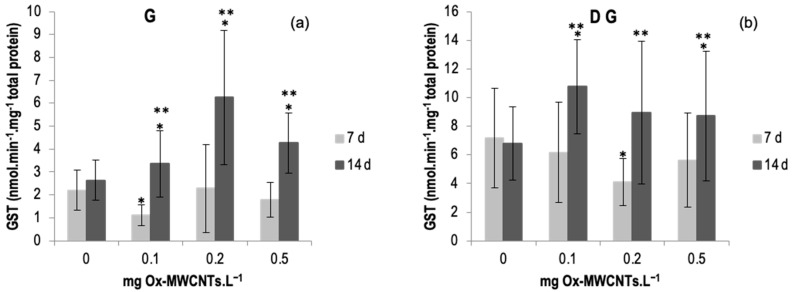
(**a**) GST activities measured in gills and (**b**) in digestive glands of *C. fluminea* exposed to different concentrations of Ox-MWCNTs. Legend: significant differences from controls (*) and between exposure periods (**).

**Figure 5 ijms-24-16122-f005:**
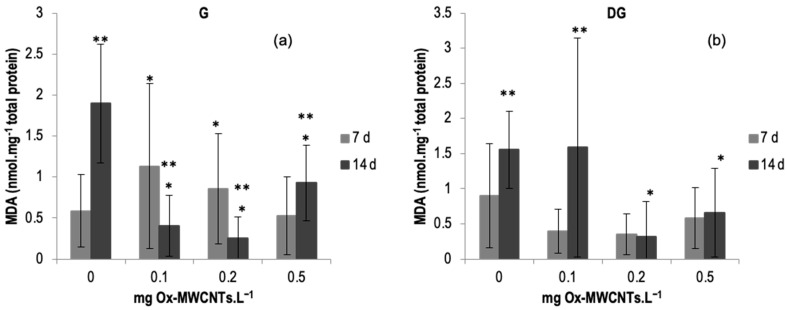
(**a**) MDA contents measured in gills and (**b**) in digestive glands of *C. fluminea* exposed to different concentrations of Ox-MWCNTs. Legend: significant differences from controls (*) and between exposure periods (**).

**Figure 6 ijms-24-16122-f006:**
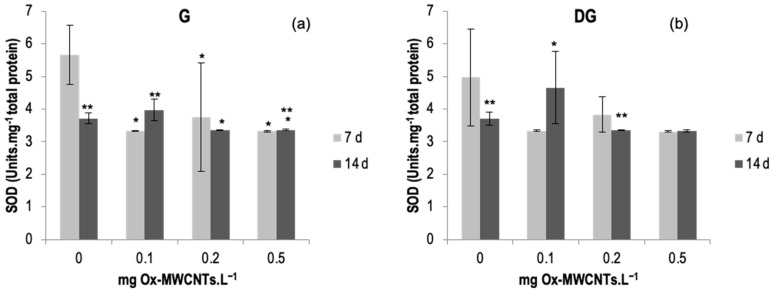
(**a**) SOD activities measured in gills and (**b**) in digestive glands of *C. fluminea* exposed to different concentrations of Ox-MWCNTs. Legend: significant differences from controls (*) and between exposure periods (**).

**Figure 7 ijms-24-16122-f007:**
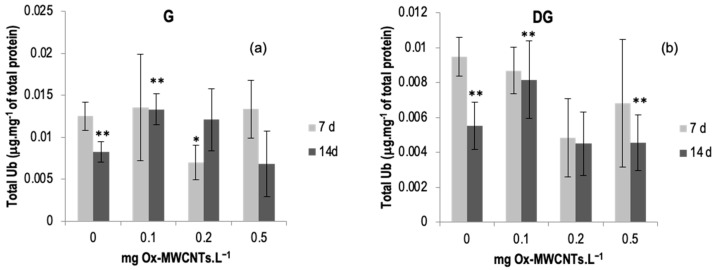
(**a**) Total ubiquitin measured in gills and (**b**) in digestive glands of *C. fluminea* exposed to different concentrations of Ox-MWCNTs. Legend: significant differences from controls (*) and between exposure periods (**).

**Figure 8 ijms-24-16122-f008:**
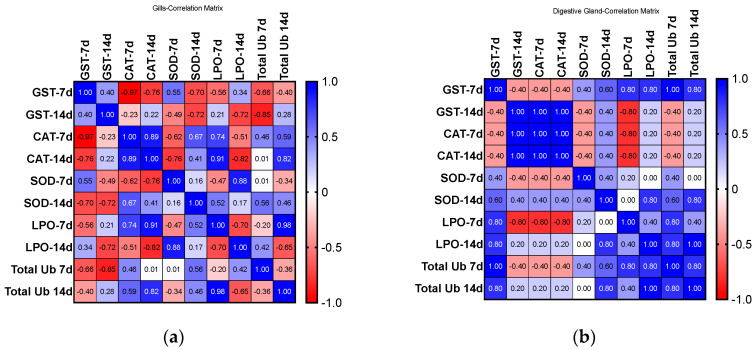
(**a**) Correlation matrix for G and (**b**) DG (values: Pearson r correlation coefficient).

## Data Availability

The data that support the findings of this study are available from the corresponding author, A.C.-S., upon reasonable request.

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
