# Peer review of "Exposure to Oxidized Multi-Walled CNTs Can Lead to Oxidative Stress in the Asian Freshwater Clam *Corbicula fluminea* (Müller, 1774)"

_ijms, 2023, doi:10.3390/ijms242216122_

Round 1
Reviewer 1 Report
Comments and Suggestions for Authors
In this study, the effect of oxidized multi-walled carbon nanotubes in the Asian freshwater clam Corbicula fluminea are investigated. I think the authors present interesting data. However, there are some points that need to be corrected. They are listed below.
Whole text, C. fluminea is shown in italics.
Title, At first it would be better not to abbreviate carbon nanotubes as CNS.
"(Müller, 1774)" is not necessary. How about a little more explanation about the bivalves used in the introduction section?
Introduction, The reason for the concentration of Ox-MWCNTs selected for this study should be stated in the Introduction, not in the Discussion.
Line 97, "C. fluminea" appears for the first time in the text, it should be corrected to "Corbicula fluminea".
Lines 102 and 107, It should be corrected to "C. fluminea".
Fig. 1, What do the two peaks indicate? What do D and G mean respectively?
Line 130, Where is the zeta potential shown?
Line 136-138, Data should be presented in figures and tables. It is difficult to understand.
Line 231, It is necessary to explain in advance why total ubiquitin was measured in this experiment.
Figs.3~7, In the text, digestive gland and gills are indicated as DG and G, respectively. They should be shown similarly in the figures.
P-value should be indicated in the legend.
Fig.6, The author indicates SOD content, but isn't that a mistake for SOD activity?
Author Response
Thank you very much for your kind words about our work.
- Whole text, C. fluminea is shown in italics.
As far as we know, the scientific names of all species should appear in italics or underlined.
- Title, At first it would be better not to abbreviate carbon nanotubes as CNS.
At the reviewer's suggestion, we have removed the CNTs abbreviation from the abstract. We believe that doing so in the title it would make the title too long.
- "(Müller, 1774)" is not necessary. How about a little more explanation about the bivalves used in the introduction section?
We believe that the most correct scientific name for the Asian clam should include the name of its discoverer.
Regarding the bivalves used, their utilization is more than validated in numerous scientific studies and recognized by the OECD as an invasive species, including their commercial use and for human consumption, which is where the importance of these filtering biomodels as markers of aquatic contaminants lies. They are also indicators of consumption and environmental health.
- Introduction, The reason for the concentration of Ox-MWCNTs selected for this study should be stated in the Introduction, not in the Discussion.
Following the reviewer's advice, we have changed the explanation of the CNT concentrations selected for this study from the discussion section to the introduction section.
- Line 97, "C. fluminea" appears for the first time in the text, it should be corrected to "Corbicula fluminea".
The text has been changed following the reviewer's suggestion.
- Lines 102 and 107, It should be corrected to "C. fluminea".
The text has been changed following the reviewer's suggestion.
- 1, What do the two peaks indicate? What do D and G mean respectively?
The D band is a characteristic band representing the defect in the graphene sheets, while the G band is one of the vibrations of the sp2-hybridized carbon atoms in the graphene sheets.
- Line 130, Where is the zeta potential shown?
The text was updated to correct it.
- Line 136-138, Data should be presented in figures and tables. It is difficult to understand.
Following the reviewer’s suggestion, tables with data are included, but as supplementary material to avoid unnecessary lengthening of the manuscript.
- Line 231, It is necessary to explain in advance why total ubiquitin was measured in this experiment.
The total ubiquitin (Ub) assay was carried out to evaluate the protein quality and integrity since the increase of ROS can lead to the formation of malfunctioning oxidized proteins, which become more susceptible to removal and degradation. Furthermore, ubiquitin-dependent proteasomal degradation machinery is sensitive to oxidative stress, beginning to decline its activity in high H2O2 concentrations. Therefore, under oxidative stress, it is expected an increase in the total ubiquitin protein to signal damaged proteins for degradation. It was also explained in the discussion section.
- 3~7, In the text, digestive gland and gills are indicated as DG and G, respectively. They should be shown similarly in the figures. P-value should be indicated in the legend.
The figures' legends are updated following the reviewer’s queries.
- 6, The author indicates SOD content, but isn't that a mistake for SOD activity?
It is a mistake we corrected to activity.
Reviewer 2 Report
Comments and Suggestions for Authors
The manuscript successfully investigates the toxicity of Ox-MWCNTs using the freshwater bivalve (Corbicula fluminea). The article is easy to read, references both in the introduction and in the discussion are appropriate. The proposed research is well designed, and the manuscript is accordingly well organized.
While the research approach and findings offer a valuable contribution to the field, there are certain areas where clarity, consistency, and depth could be enhanced.
There are some typos to be corrected.
I would suggest that the authors define the used acronyms. There are many undefined acronyms when first used. This helps readers understand the meaning of these terms.
Scale bars are blurred in some subfigures.
Section 4 should be inserted immediately after the Introduction.
line 414: Citation is needed in "Habig et al., 1974".
line 445: Citation is needed in "McCord & Fridovich, 1969"
At first glance, the conclusions lack quantitative results.
There are some language modifications to simplify and clarify the content for the reader.
The discussion section: Some correlations of the results with the literature are recommended.
The results of statistical analysis have not been discussed in detail.
The Introduction section: Blocks References, e.g. “[20–27]”, “[26,27,32,33]” should be avoided as these do not emphasize the particular aspects from the cited papers. Particularly, when citations are made in reference to specific technical aspects, single/double, e.g. [1, 2] references are encouraged. It is strongly suggested that the references need to make in-depth comments on the content of the each cited paper; avoid generic comments.
There are some language modifications to simplify and clarify the content for the reader.
Author Response
Thank you very much for your kind words about our work.
- I would suggest that the authors define the used acronyms. There are many undefined acronyms when first used. This helps readers understand the meaning of these terms.
The text has been updated so that each time an abbreviation appears, it is defined.
- Scale bars are blurred in some subfigures.
The legends have been updated to clarify the scale of the bars for the TEM and HRTEM micrographs.
- Section 4 should be inserted immediately after the Introduction.
Following the journal's template and instructions, we have placed the materials and methods section in the section assigned by the journal for that purpose.
- line 414: Citation is needed in "Habig et al., 1974".
- line 445: Citation is needed in "McCord & Fridovich, 1969"
The two studies in the previous references are already in the right place.
- At first glance, the conclusions lack quantitative results.
Dear reviewer, the quantitative results allow us to draw conclusions. More experiments will be necessary to extract more information.
- There are some language modifications to simplify and clarify the content for the reader.
- The discussion section: Some correlations of the results with the literature are recommended.
- The results of statistical analysis have not been discussed in detail.
We have tried to address these three reviewer suggestions as much as possible.
- The Introduction section: Blocks References, e.g. “[20–27]”, “[26,27,32,33]” should be avoided as these do not emphasize the particular aspects from the cited papers. Particularly, when citations are made in reference to specific technical aspects, single/double, e.g. [1, 2] references are encouraged. It is strongly suggested that the references need to make in-depth comments on the content of the each cited paper; avoid generic comments.
The reference blocks have been minimized as much as possible following the reviewers' instructions.
Reviewer 3 Report
Comments and Suggestions for Authors
This study investigated whether exposure, even at lower concentrations, will produce changes in antioxidant defenses and cellular damage biomarkers, suggesting that Ox-MWCNTs may cause oxidative stress and cell damage.
Major issues
Please describe in detail the procedure for collection of the bivalves. How did you choose this particular site? How was the technique for collection? How did you choose individuals? Did you have any procedures for inclusion and exclusion criteria for the individuals?
Analysis. Did you carry out any Dunn’s comparisons? Also, if the data were not normally distributed, why carry out Pearson’s correlation analysis and not Spearman rank? If necessary, please redo the correlation analysis means of Spearman rank correlation analysis.
Lack of tables in the results makes arduous comprehension of the findings. Graphs are fine and very helpful but do not provide the details that can be seen in tables. Hence, please add tables in all sub-sections with summaries of results.
Some recent relevant references are missing.
Please add a new passage in Discussion with the clinical significance of the findings for aquatic health management.
Minor issues
The Introduction is rather longer than necessary, so I suggest that some passages are deleted and possibly transferred to the Discussion, where they could be useful.
I suggest to separate the Discussion in two sub-sections for easier flow of reading.
Author Response
Thank you very much for your kind words about our work.
Major issues
- Please describe in detail the procedure for collection of the bivalves. How did you choose this particular site? How was the technique for collection? How did you choose individuals? Did you have any procedures for inclusion and exclusion criteria for the individuals?
The selection of the collection site was based on several issues: 1. the distance from ports and other points of high human contamination, 2. confirmation that Corbicula fluminea is present, and 3. the distance to travel with the individuals once they have been collected.
The collection technique seeks to produce less stress for the individuals, so it was carried out when the tide was low, a suitable container was filled with water from the same environment in which the collection was carried out and the selection was carried out. adult individuals and approximately the same shell size. Discarding individuals that may be much larger or smaller than average. Obviously, individuals with damaged shells were quickly discarded because they were under stressful conditions.
- Did you carry out any Dunn’s comparisons? Also, if the data were not normally distributed, why carry out Pearson’s correlation analysis and not Spearman rank? If necessary, please redo the correlation analysis means of Spearman rank correlation analysis.
Dunn’s comparison is similar to that of ANOVA treatment; in our statistic section, we carried out the Kruskal-Wallis test to identify significant differences between controls and treatments. A significance level was fixed at 5%. Spearman rank test serves only to establish the correlation between two variables, so for relating variables, we selected a more suitable method as is Correlation analysis.
- Lack of tables in the results makes arduous comprehension of the findings. Graphs are fine and very helpful but do not provide the details that can be seen in tables. Hence, please add tables in all sub-sections with summaries of results.
Following the reviewer’s suggestion, tables with data are included, but as supplementary material to avoid unnecessary lengthening of the manuscript.
- Some recent relevant references are missing.
We have already added some recent references.
- Please add a new passage in Discussion with the clinical significance of the findings for aquatic health management.
When discussing the effects on aquatic biota, we cannot speak about clinical significance, a term related to human health. However, the results provide information about the ecosystem's health, including aquatic biota, if contamination with CNTs occurs. Nonetheless, if these bivalves are consumed by humans, then some effects can occur, mainly in lung epithelial cells (Lucas et al., 2021) but also in the cardiovascular system (Long et al., 2017) or the immune system (Li et al., 2018) among other effects, however, the knowledge on its toxicity determinants is still to clarify (Allegri et al., 2016) and thus a better understanding is needed (Zhou et al., 2017).
Minor issues
- The Introduction is rather longer than necessary, so I suggest that some passages are deleted and possibly transferred to the Discussion, where they could be useful.
We will follow the suggestion as much as possible.
- I suggest to separate the Discussion in two sub-sections for easier flow of reading.
The authors followed the guidelines of the journal in terms of manuscript sections.
Round 2
Reviewer 3 Report
Comments and Suggestions for Authors
The authors have improved the manuscript, by taking into account all the comments made and by making appropriate corrections as indicated.
I have now further comments.